# AUTOMATIC MUSIC PRODUCTION USING GENERATIVE ADVERSARIAL NETWORKS

## ABSTRACT

When talking about computer-based music generation, two are the main threads of research: the construction of *autonomous music-making systems*, and the design of *computer-based environments to assist musicians*. However, even though creating accompaniments for melodies is an essential part of every producer's and songwriter's work, little effort has been done in the field of automatic music arrangement in the audio domain. In this contribution, we propose a novel framework for *automatic music accompaniment in the Mel-frequency domain*. Using several songs converted into Mel-spectrograms – a two-dimensional time-frequency representation of audio signals – we were able to automatically generate original arrangements for both bass and voice lines. Treating music pieces as images (Mel-spectrograms) allowed us to reformulate our problem as an *unpaired image-to-image translation* problem, and to tackle it with CycleGAN, a well-established framework. Moreover, the choice to deploy raw audio and Mel-spectrograms enabled us to more effectively model long-range dependencies, to better represent how humans perceive music, and to potentially draw sounds for new arrangements from the vast collection of music recordings accumulated in the last century. Our approach was tested on two different downstream tasks: given a bass line creating credible and on-time drums, and given an acapella song arranging it to a full song. In absence of an objective way of evaluating the output of music generative systems, we also defined a possible metric for the proposed task, partially based on human (and expert) judgement.

## 1 INTRODUCTION

The development of home music production has brought significant innovations into the process of pop music composition. Software like Pro Tools, Cubase, and Logic – as well as MIDI-based technologies and digital instruments – provide a wide set of tools to manipulate recordings and simplify the composition process for artists and producers. After recording a melody, maybe with the aid of a guitar or a piano, song writers can now start building up the arrangement one piece at a time, sometimes not even needing professional musicians or proper music training. As a result, singers and song writers – as well as producers – have started asking for tools that could facilitate, or to some extent even automate, the creation of full songs around their lyrics and melodies. To meet this new demand, the goal of designing computer-based environments to assist human musicians has become central in the field of automatic music generation (Briot et al., 2020). IRCAM OpenMusic (Assayag et al., 1999), Sony CSL-Paris FlowComposer (Papadopoulos et al., 2016), and Logic Pro X Easy Drummer are just some examples. In addition, more solutions based on deep learning techniques, such as RL-Duet (Jiang et al., 2020) – a deep reinforcement learning algorithm for online accompaniment generation – or PopMAG, a transformer-based architecture which relies on a multi-track MIDI representation of music (Ren et al., 2020), continue to be studied. A comprehensive review of the most relevant deep learning techniques applied to music is provided by (Briot et al., 2020). Most of these strategies, however, suffer from the same critical issue, which makes them less appealing in view of music production for commercial purposes: they rely on a symbolic/MIDI representation of music. The approach proposed in this paper, instead, is a first attempt at automatically generating an euphonic arrangement (two or more sound patterns that produce a pleasing and harmonious piece of music) in the audio domain, given a musical sample encoded in a two-dimensional time-frequency representation (in particular, we opted for the Mel-spectrogram time-frequency representation). Al-

though arrangement generation has been studied in the context of symbolic audio, indeed, switching to Mel-spectrograms allows us to preserve the sound heritage of other musical pieces (allowing operations such as sampling) and is more suitable for real-life cases, where voice, for instance, cannot be encoded in MIDI.

We focused our attention on two different tasks of increasing difficulty: (i) given a bass line to create credible and on-time drums, and (ii) given the voice line, to output a new and euphonic musical arrangement. Incidentally, we found out that – for training samples – our model was able to reconstruct the original arrangement pretty well, even though no pairing among the Mel-spectrograms of the two domains was performed. By means of the Mel-spectrogram representation of music, we can consider the problem of automatically generating an arrangement or accompaniment for a specific musical sample equivalent to an image-to-image translation task. For instance, if we have the Mel-spectrogram of an acapella song, we may want to produce the Mel-spectrogram of the same song including a suitable arrangement. To solve this task, we tested an unpaired image-to-image translation strategy known as CycleGAN (Zhu et al., 2017), which consists of translating an image from a source domain X to a target domain Y in the absence of paired examples, by training both the mapping from X to Y and from Y to X simultaneously, with the goal of minimizing a cycle consistency loss. The aforementioned system was trained on 5s pop music samples (equivalent to $256 \times 256$ Mel-spectrograms) coming both from the Free Music Archive (FMA) dataset (Defferrard et al., 2017; 2018), and from the Demucs dataset (Défossez et al., 2019). The short sample duration does not affect the proposed methodology, at least with respect to the arrangement task we focus on, and inference can be performed also on full songs. Part of the dataset was pre-processed first, since the FMA songs lack source separated channels (i.e. differentiated vocals, bass, drums, etc.). The required channels were extracted using Demucs (Défossez et al., 2019). The main innovations presented in this contribution are as follows: (i.) treating music pieces as images, we developed a framework to automatically generate music arrangement in the Mel-frequency domain, different from any other previous approach; (ii.) our approach is able to generate arrangements with low computational resources and limited inference time, if compared to other popular solutions for automatic music generation (Dhariwal et al., 2020); (iii.) we developed a metric – partially based on or correlated to human (and expert) judgement – to automatically evaluate the obtained results and the creativity of the proposed system, given the challenges of a quantitative assessment of music. To the best of our knowledge, this is the first work to face the automatic arrangement production task in the audio domain by leveraging a two-dimensional time-frequency representation.

## 2 RELATED WORKS

The interest surrounding automatic music generation, translation and arrangement has greatly increased in the last few years, as proven by the high numbers of solutions proposed – see (Briot et al., 2020) for a comprehensive and detailed survey. Here we present a brief overview of the key contributions both in symbolic and audio domain.

**Music generation & arrangement in the symbolic domain.** There is a very large body of research that uses a symbolic representation of music to perform music generation and arrangement. The following contributions used MIDI, piano rolls, chord and note names to feed several deep learning architectures and tackle different aspects of the music generation problem. In (Yang et al., 2017), CNNs are used for generating melody as a series of MIDI notes either from scratch, by following a chord sequence, or by conditioning on the melody of previous bars. In (Mangal et al., 2019; Jaques et al., 2016; Mogren, 2016; Makris et al., 2017), LSTM networks are used to generate musical notes, melodies, polyphonic music pieces, and long drum sequences, under constraints imposed by metrical rhythm information and a given bass sequence. The authors of (Yamshchikov & Tikhonov, 2017; Roberts et al., 2018), instead, use VAE networks to generate melodies. In (Boulanger-Lewandowski et al., 2012), symbolic sequences of polyphonic music are modeled in a completely general piano-roll representation, while the authors of (Hadjeres & Nielsen, 2017) propose a novel architecture to generate melodies satisfying positional constraints in the style of the soprano parts of the J.S. Bach chorale harmonisations encoded in MIDI. In (Johnson, 2017), RNNs are used for prediction and composition of polyphonic music; in (Hadjeres et al., 2017), highly convincing chorales in the style of Bach were automatically generated using note names; (Lattner et al., 2018) added higher-level structure on generated, polyphonic music, whereas (Mao et al., 2018) designed an end-to-end generative model capable of composing music conditioned on a specific mixture of composer styles. The

approach described in (Hawthorne et al., 2018), instead, relies on notes as an intermediate representation to a suite of models – namely, a transcription model based on a CNN and a RNN network (Hawthorne et al., 2017), a self-attention-based music language model (Huang et al., 2018) and a WaveNet model (Oord et al., 2016) – capable of transcribing, composing, and synthesizing audio waveforms. Finally, (Zhu et al., 2018) proposes an end-to-end melody and arrangement generation framework, called XiaoIce Band, which generates a melody track with several accompaniments played by several types of instruments. As this extensive literature on music generation in the symbolic domain shows, a promising approach would be to work with symbolic music and then use state-of-the-art synthesizers to produce sounds. MIDI, music sheets and piano rolls, however, are not always easy to find or produce. Moreover, many musicians and artists can not read music and would be more comfortable to work in a less formalized setting. Finally, state-of-the-art synthesizers, although increasingly indistinguishable from live recordings, can not yet reproduce the infinite nuances of real voices and instruments. Conversely, raw audio representation could be more appealing for some creators given its flexibility and little music competence required.

**Music generation & arrangement in the audio domain.** Some of the most relevant approaches proposed so far in the field of waveform music generation deal with raw audio representation in the time domain. Many of these approaches draw methods and ideas from the extensive literature on audio and speech synthesis. For instance, in (Prenger et al., 2019) a flow-based network capable of generating high quality speech from mel-spectrograms is proposed, while in (Wang et al., 2019) the authors present a neural source-filter (NSF) waveform modeling framework that is straightforward to train and fast to generate waveforms. In (Zhao et al., 2020) recent neural waveform synthesizers such as WaveNet, WaveG-low, and the neural-source-filter (NSF) models are compared. (Mehri et al., 2016) tested a model for unconditional audio generation based on generating one audio sample at a time, and (Bhave et al., 2019) applied Restricted Boltzmann Machine and LSTM architectures to raw audio files in the frequency domain in order to generate music. A fully probabilistic and autoregressive model, with the predictive distribution for each audio sample conditioned on all previous ones, is used in (Oord et al., 2016) to produce novel and often highly realistic musical fragments. (Manzelli et al., 2018) combined two types of music generation models, namely symbolic and raw audio models, to train a raw audio model based on the WaveNet architecture, but that incorporates the notes of the composition as a secondary input to the network. Finally, in (Dhariwal et al., 2020) the authors tackled the long context of raw audio using a multi-scale VQ-VAE to compress it to discrete codes, and modeled such context through Sparse Transformers, in order to generate music with singing in the raw audio domain. Nonetheless, due to the computational resources required to directly model long-range dependencies in the time domain, either short samples of music can be generated or complex and large architectures and long inference time are required. On the other hand, in (Vasquez & Lewis, 2019) a novel approach is discussed, which proves that long-range dependencies can be more tractably modelled in two-dimensional time-frequency representations such as Mel-spectrograms. More precisely, the authors of this contribution designed a highly expressive probabilistic model and a multiscale generation procedure over Mel-spectrograms capable of generating high-fidelity audio samples which capture structure at timescales. It is worth recalling, as well, that treating spectrograms as images is the current standard for many Music Information Retrieval tasks, such as music transcription (Sigtia et al., 2016) and chord recognition.

**Generative adversarial networks for music generation.** Our work is precisely founded on this novel assumption, thus taking the best from the raw audio representation, while tackling the main issues induced by musical signals long-range dependencies thanks to the waveform-to-spectrograms conversion. Such two-dimensional representation of music paves the way to the application of several image processing techniques and image-to-image translation networks to carry out style transfer and arrangement generation (Isola et al., 2017; Zhu et al., 2017). It is worth recalling that the application of GANs to music generation tasks is not new: in (Brunner et al., 2018), Generative Adversarial Networks are applied on symbolic music to perform music genre transfer; however, to the best of our knowledge, GANs have never been applied to raw audio in the Mel-frequency domain for music generation purposes. As to the arrangement generation task, also in this case the large majority of approaches proposed in literature is based on symbolic representation of music: in (Ren et al., 2020), a novel Multi-track MIDI representation (MuMIDI) is presented, which enables simultaneous multi-track generation in a single sequence and explicitly models the dependency of the notes from different tracks by means of a Transformer-based architecture; in (Jiang et al., 2020), a deep reinforcement learning algorithm for online accompaniment generation is described. Coming to the most relevant issues in the development of music generation systems, both the training and

evaluation of such systems haven proven challenging, mainly because of the following reasons: (i) the available datasets for music generation tasks are challenging due to their inherent high-entropy (Dieleman et al., 2018), and (ii) the definition of an objective metric and loss is a common problem to generative models such as GANs: at now, generative models in the music domain are evaluated based on the subjective response of a pool of listeners, and just for the MIDI representation a set of simple musically informed objective metrics was proposed (Yang & Lerch, 2020).

## 3 METHOD

### 3.1 SOURCE SEPARATION FOR MUSIC

We present a novel framework for automatic music arrangement generation using an adversarially trained deep learning model. A key challenge to our approach is the scarce availability of music data featuring source separated channels (i.e. differentiated vocals, bass, drums, ...). To this end, we leverage Demucs by Défossez et al., a freely available tool, which separates music into its generating sources. Demucs features a U-NET encoder-decoder architecture with a bidirectional LSTM as middle hidden layer. In particular we used a pre-trained model made available by the original authors, consisting of 6 convolutional encoder and decoder blocks and a middle hidden size of length 3200. Demucs is time–equivariant, meaning that shifts in the input mixture will cause a congruent shifts in the output. The model does not feature this property naturally, but it is achieved through a workaround (randomized equivariant stabilization) as explained by the original authors. Nonetheless, at times this method produces noisy separations – with watered-down harmonics and traces of other instruments in the vocal segment – effectively hindering the ability of later part of the pipeline to properly recognise and reconstruct accompaniment, of which harmonics are a critical part. While better source-separation methods are available [Sota SDR = 5.85, Demucs SDR = 5.67], we chose to use Demucs because it was faster and easier to embed in our pipeline. Moreover, for bass source separation it beats the state of the art [Sota SDR = 5.28, Demucs SDR = 6.21]. Finally, we were adamant about picking a tool using deep learning because this may open the possibility to build an end-to-end trained pipeline in the future. This at least partially solves the challenge of data availability and allows us to feed our model with the appropriate signals.

### 3.2 MUSIC REPRESENTATION – FROM RAW AUDIO TO MEL-SPECTROGRAMS

One of the main features of our method is to choose a two-dimensional time-frequency representation of the audio samples rather than a time representation. The spectrum is a common transformed representation for audio, obtained via a Short-Time Fourier transform (STFT). The discrete STFT of a given signal $x : [0 : L-1] := \{0, 1, \ldots, L-1\} \to \mathbb{R}$ leads to the $k^{\text{th}}$ complex Fourier coefficient for the $m^{\text{th}}$ time frame $\mathcal{X}(m, k) := \sum_{n=0}^{N-1} x(n + mH) \cdot w(n) \cdot e^{-\frac{2\pi i k n}{N}}$, with $m \in [0 : M]$ and $K \in [0 : K]$, and where $w(n)$ is a sampled window function of length $N \in \mathbb{N}$ and $H \in \mathbb{N}$ is the hop size, which determines the step size in which the window is to be shifted across the signal (Müller, 2015). The spectrogram is a two-dimensional representation of the squared magnitude of the STFT, i.e. $\mathcal{Y}(m, k) := |\mathcal{X}(m, k)|^2$, with $m \in [0 : M]$ and $K \in [0 : K]$. Figure 1 shows a Mel-spectrogram example (Stevens et al., 1937), which is treated as single channel image, representing the sound intensity with respect to time – x axis – and frequency – y axis (Briot et al., 2020). This decision allows to better deal with long-range dependencies typical of such kind of data and to reduce the computational resources and inference time required. Moreover, the Mel-scale is based on a mapping between the actual frequency $f$ and perceived pitch $m = 2595 \cdot log_{10}(1 + \frac{f}{700})$, as the human auditory system does not perceive pitch in a linear manner. Finally, using Mel-spectrograms of pre-existing songs to train our model potentially enables to draw sounds for new arrangements from the vast collection of music recordings accumulated in the last century.

After the source separation task was carried out on our song dataset, each source (and the full song) waveforms were turned into corresponding Mel-spectrograms. This has been done using PyTorch Audio[1], to take advantage of robust, GPU accelerated conversion. We decided to discard the phase information in this process, to reduce the dimensionality of the representation. To revert back to the time-domain signal, we: (i.) apply a conversion matrix (using triangular filter banks)

---

[1]Available at: https://pytorch.org/audio/stable/index.html

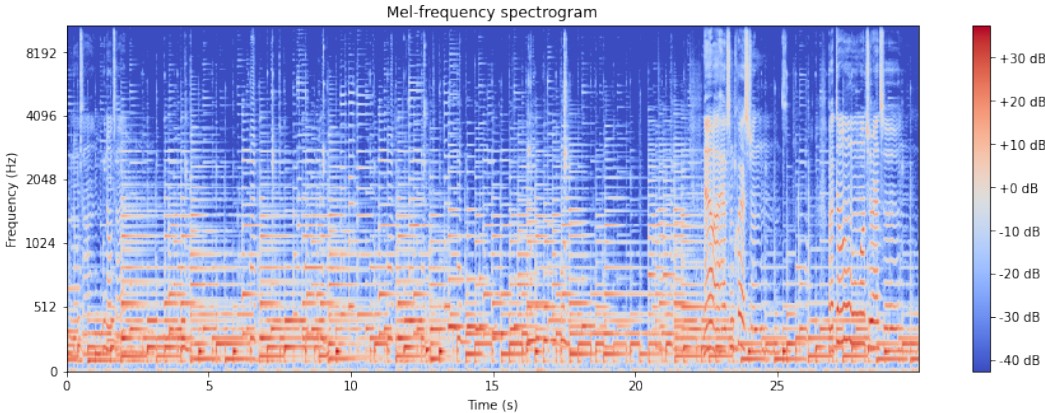

Figure 1: Example of a Mel-spectrogram

to convert the Mel-frequency STFT to a linear scale STFT, where the matrix is calculated using a gradient-based method (Decorsière et al., 2015) to minimize the euclidean norm between the original Mel-spectrogram and the product between reconstructed spectrogram and filter banks; (ii.) use the Griffin-Lim's algorithm (Griffin & Jae Lim, 1984) to reconstruct phase information.

It is worth noticing that Mel-scale conversion and the removal of STFT phases respectively discard frequency and temporal information, that results in a distortion in the recovered signal. To minimize this problem, we use a high-enough resolution of the Mel-spectrograms (Vasquez & Lewis, 2019), whose size can be tweaked with *number of mels* and *STFT hop size* parameters. Thus, the optimal parameters we found were the following ones: the sampling rate was initially set to 22050 Hz, the window length $N$ to 2048, the number of Mel-frequency bins to 256 and the hop size $H$ to 512. To fit our model requirements, we cropped out $256 \times 256$ windows from each Mel-spectrogram with an overlapping of 50 time frames, obtaining multiple samples from each song (each equivalent to 5 seconds of music).

### 3.3 IMAGE TO IMAGE TRANSLATION - CYCLEGAN

The automatic arrangement generation task was faced through an *unpaired image-to-image translation* framework, by adapting the CycleGAN model to our purpose. CycleGAN is a framework able to translate between domains without paired input-output examples, by assuming some underlying relationship between the domains and trying to learn that relationship. Based on a set of images in domain $X$ and a different set in domain $Y$, the algorithm learns both a mapping $G : X \to Y$ and a mapping $F : Y \to X$, such that the output $\hat{y} = G(x)$ for every $x \in X$, is indistinguishable from images $y \in Y$ and $\hat{x} = G(y)$ for every $y \in Y$, is indistinguishable from images $x \in X$. The other relevant assumption is that, given a mapping $G : X \to Y$ and another mapping $F : Y \to X$, then $G$ and $F$ should be inverses of each other, and both mappings should be bijections. This assumption is implemented by training both the mapping $G$ and $F$ simultaneously, and adding a cycle consistency loss that encourages $F(G(x)) \approx x$ and $G(F(y)) \approx y$. The cycle consistency loss is then combined with the adversarial losses on domains $X$ and $Y$ (Zhu et al., 2017).

### 3.4 AUTOMATIC MUSIC PRODUCTION

The method we propose takes as input a set of $N$ music songs in the waveform domain $X = \{\mathbf{x_i}\}_{i=1}^N$, where $\mathbf{x_i}$ is a waveform whose number of samples depends on the sampling rate and the audio length. Each waveform is then separated by Demucs into three different sources. Thus, we end up having four different WAV files for each song, which means a new set of data of the kind: $X_{\text{NEW}} = \{\mathbf{x_i}, \mathbf{v_i}, \mathbf{d_i}, \mathbf{b_i}\}_{i=1}^N$, where $\mathbf{v_i}, \mathbf{b_i}, \mathbf{d_i}$ represents vocal, bass, and drums respectively. Each track is then converted to its Mel-spectrogram representation. Since the CycleGAN model takes $256 \times 256$ images as input, each spectrogram is chunked into smaller pieces with an overlapping window of 50 time frames; finally, in order to obtain one channel images from the original spectro-

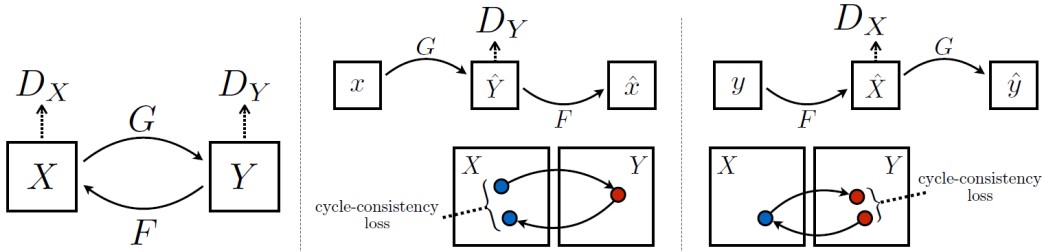

Figure 2: Representation of the CycleGAN model, which consists of two mapping functions $G$ and $F$, two discriminators $D_X$ and $D_Y$ and two cycle-consistency losses (Zhu et al., 2017)

grams, we performed a discretization step in the range $[0 - 255]$. In the final stage of our pipeline, we feed the obtained dataset to the CycleGan model, that has been adapted to the structure of this data. Even though the discretization step introduces some distortion – original spectrogram values are floats – the impact on the audio quality is negligible.

At training time, as the model takes into account two domains $X$ and $Y$, we considered two different experimental settings of increasing difficulty: (i.) we feed the model with bass and drums lines in order to create suitable drums given a bass line; (ii.) we take the vocals and the whole song respectively, with the goal of generating an arrangement euphonic to the vocal line. Solving the bass2drums task effectively would represent a first interesting intermediary goal. Drums and bass are usually the first instruments to be recorded when producing a song. Ideally, though, a system the automatically generates a full song given a voice line as input would be far more ambitious and disruptive because it would allow anyone to express her/himself in music.

## 4 EXPERIMENTS

### 4.1 DATASET

For the quality of the generated music samples, it is important to carefully pick the training dataset. To train and test our model we decided to use the Free Music Archive (FMA), and the musdb18 dataset (Rafii et al., 2017) that were both made available quite recently. The Free Music Archive (FMA) is the largest publicly available dataset suitable for music information retrieval tasks (Defferrard et al., 2017; 2018). In its full form it provides 917 GB and 343 days of Creative Commons-licensed audio from 106,574 tracks from 16,341 artists and 14,854 albums, arranged in a hierarchical taxonomy of 161 unbalanced genres. It provides full-length and high-quality audio, pre-computed features, together with track- and user-level metadata, tags, and free-form text such as biographies. Given the size of FMA, we chose to select only untrimmed songs tagged as either pop, soul-RnB, or indie-rock, for a total of approximately 10,000 songs ( 700 hours of audio). It is possible to read the full list of songs at **FMA website**, selecting the genre. We discarded all songs that were recorded live by filtering out all albums that contained the word "live" in the title. Finally, in order to better validate and fine-tune our model we decided to also use the full musdb18 dataset. This rather small dataset is made up of 100 tracks taken from the DSD100 dataset, 46 tracks from the MedleyDB, 2 tracks kindly provided by Native Instruments, and 2 tracks from the Canadian rock band The Easton Ellises. It represents a unique and precious source of songs delivered in multi-track fashion. Each song comes as 5 audio files – vocals, bass, drums, others, full song – perfectly separated at the master level. We used the 100 tracks taken from the DSD100 dataset to fine-tune the model ( 6.5 hours), and the remaining 50 songs to test it ( 3.5 hours). It is worth noting that DEMUCS is not a perfect method for source separation: it introduces artefacts and noise of the original song in the separated sources output, making the task easier and inducing the model to learn to amplify it. For this reason our training strategy is to pre-train with the artificial FMA dataset then fine-tune with musdb18. Intuitively, the former, which is much larger, helps the model to create a good representation of musical signal; the latter, which is of higher quality, contributes to reducing bias induced by the noise and to further specializing to generate a base relying only on the (clean) input given.

Table 1: Pearson's correlation matrix for all 4 annotators

|  | Guitarist | Drummer | Producer 1 | Producer 2 |
|---|---|---|---|---|
| **Guitarist** | na | 0.82 | 0.75 | 0.77 |
| **Drummer** | 0.82 | na | 0.76 | 0.79 |
| **Producer 1** | 0.75 | 0.76 | na | 0.85 |
| **Producer 2** | 0.77 | 0.79 | 0.85 | na |

## 4.2 TRAINING OF THE CYCLE-GAN MODEL

For both the bass2drums and voice2song tasks, we trained our model on 2 Tesla V100 SXM2 GPUs with 32 GB of RAM for 12 epochs (FMA dataset), and fine-tuned it for 20 more epochs (musdb18 dataset). Each task required 6 days of training. For both the settings, as a final step, the spectrograms obtained were converted to the waveform domain, to evaluate the produced music. As to the Cycle-GAN model used for training, we relied on the default network available at **this GitHub repository**. As a result, the model uses a **resnet_9blocks** ResNet generator and a basic 70x70 PatchGAN as a discriminator. The Adam optimizer (Kingma & Ba, 2014) was chosen both for the generators and the discriminators, with betas $(0.5, 0.999)$ and learning rate equal to 0.0002. The batch size was set to 1. The $\lambda$ weights for cycle losses were both equal to 10.

## 4.3 EXPERIMENTAL SETTING

There is an intrinsic difficulty in objectively evaluating artistic artifacts such as music. As a human construct, there are no objective, universal criteria for appreciating music. Nevertheless, in order to establish some form of benchmark and allow comparisons among different approaches, many generative approaches to raw audio, such as Jukebox (Dhariwal et al., 2020) or Universal Music Translation Network (Mor et al., 2018), try to overcome this obstacle by having the results manually tagged by human experts. Although this rating may be the best in terms of quality, the result is still somehow subjective, thus different people may end up giving different or biased ratings based on their personal taste. Moreover, the computational cost and time required to manually annotate the dataset could become prohibitive even for relatively few samples (over 1000). Aware of the great limits linked to this human-based approach and unable to find a more convincing evaluation procedure, we propose a new metric that highly correlates with human judgment. This could represent a first benchmark for the tasks at hand. The results remain somehow subjective, but at least we were able to automatically replicate our evaluators' criteria and grades, saving time and money.

## 4.4 METRICS

If we consider as a general objective for a system the capacity to assist composers and musicians, rather than to autonomously generate music, we should also consider as an evaluation criteria the satisfaction of the composer (notably, if the assistance of the computer allowed him to compose and create music that he may consider not having been possible otherwise), rather than the satisfaction of the auditors (who remain too often guided by some conformance to a current musical trend) (Briot et al., 2020).

However, as previously stated, an exclusive human evaluation may be unsustainable in terms of computational cost and time required. Thus we carried out the following quantitative assessment of our model. We first produced 400 test samples – from as many different songs and authors – of artificial arrangements and drum lines starting from voice and bass lines that were not part of the training set. We then asked a professional guitarist who has been playing in a pop-rock band for more than 10 years, a professional drum player from the same band, and two pop and indie-rock music producers with more than 4 years of experience to manually annotate these samples, capturing the following musical dimensions: quality, euphony, coherence, intelligibility. More precisely, for each sample, we asked them to rate from 1 to 10 the following aspects: (i) *Quality*: a rating from 1 to 10 of the naturalness and absence of artifacts or noise, (ii) *Contamination*: a rating from 1 to 10 of the contamination by other sources, (iii) *Credibility*: a rating from 1 to 10 of the credibility of the sample, (iv) *Time*: a rating from 1 to 10 of whether the produced drums and arrangements are on time the bass and voice lines. The choice fell on these four aspects after we asked the evaluators to

list and describe the most relevant dimensions in the perceived quality of a piece of pop-rock music. The correlation matrix for all 4 annotators is shown in Table 1.

Ideally, we want to produce some quantitative measure whose outputs – when applied to generated samples – highly correlates (i.e. predict) expert average grades. To achieve this goal, we trained a logistic regression model with features obtained through a comparison between the original arrangement and the model output, as well as the original drums and the artificial drums. Here are the details on how we obtained suitable features:

**STOI-like features.** We created a procedure – inspired by the STOI (Andersen et al., 2017) – whose output vector somehow measures the Mel-frequency bins correlation throughout time between the original sample (arrangement/drums) and the fake one. The obtained vector can then be used to feed a multi regression model whose independent variable is the human score attributed to that sample.

Here is the formalisation: $HumanScore = \sum_i^{256} a_i \left[ \sum_t^{256} (x_i^{(t)} - \bar{x}^{(t)})(y_i^{(t)} - \bar{y}^{(t)}) \right]$. To simplify, to each pair of samples (original and generated one) a 256 element long vector is associated as follows: $\mathcal{S}(\mathcal{X}, \mathcal{Y}, l)^{(i)} = \sum_t^{256} (x_i^{(t)} - \bar{x}^{(t)})(y_i^{(t)} - \bar{y}^{(t)})$. Where: (i.) $\mathcal{X}$ and $\mathcal{Y}$ are, respectively, the Mel-spectrogram matrices of original and generated samples; (ii.) $a_i$ is the $i$-th coefficient for the linear regression; (iii.) $x_i^{(t)}$ and $y_i^{(t)}$ the $i$-th element of the $t$-th column of matrices $\mathcal{X}$ and $\mathcal{Y}$, respectively; (iv.) $\bar{x}^{(t)}$ and $\bar{y}^{(t)}$ are the means along the $t$-th column of matrices $\mathcal{X}$ and $\mathcal{Y}$, respectively. Each feature $i$ of the regression model is a sort of Pearson correlation coefficient between row $i$ of $\mathcal{X}$ and row $i$ of $\mathcal{Y}$ throughout time.

**FID-based features.** In the context of GANs result evaluation, the Fréchet Inception distance (FID) is supposed to improve on the Inception Score by actually comparing the statistics of generated samples to real samples (Salimans et al., 2016; Heusel et al., 2017). In other words, FID measures the probabilistic distance between two multivariate Gaussians, where $X_r = N(\mu_r, \Sigma_r)$ and $X_g = N(\mu_g, \Sigma_g)$ are the 2048-dimensional activations of the Inception-v3 pool3 layer – for real and generated samples respectively – modeled as normal distributions. The similarity between the two distributions is measured as follow: $FID = \|\mu_r - \mu_g\|^2 + Tr(\Sigma_r + \Sigma_g - 2(\Sigma_r \Sigma_g)^{1/2})$. Nevertheless, since we want to assign a score to each sample, we just estimated the $X_r = N(\mu_r, \Sigma_r)$ parameters – using different activation layers of the Inception pre-trained network – and then we calculated the probability density associated to each fake sample. Finally, we added these scores to the regression model predictors.

## 4.5 EXPERIMENTAL RESULTS

For the bass2drums task, Figure 3 shows the distribution of grades for the 400 test samples – averaged among all four independent evaluators and over all the four dimensions. We rounded the results to the closest integer to make the plot more readable. The higher the grade, the better the sample will sound. Additionally, to fully understand what to expect from samples graded similarly, we discussed the model results with the evaluators. We collectively listened to a random set of samples and it turned out that all four raters followed similar principles in assigning the grades. Samples with grade 1-3 are generally silent or very noisy. In samples graded 4-5 few sounds start to emerge, but they are usually not very pleasant to listen to, nor coherent. Grades 6-7 identify drums that sound good, that are coherent, but that are not continuous: they tend to follow the bass line too closely. Finally, samples graded 8 and 9 are almost indistinguishable from real drums, both in terms of sound and timing. In the labeling of non graded samples phase, we therefore assigned a 0 to those samples whose average grade was between 1 and 5, and 1 to those between 6 and 10. Finally, we trained a multi-logistic regression model with both the STOI-like and the FID-based features. The model accuracy on test set was 87%.

Given this pretty good result, we could then used this trained logistic model to label 14000 different 5s fake drums clips, produced from as many real bass lines. Two third of these were labeled as good sounding and on time. **Here** is a private Sound Cloud playlist where you can listen to some of the most interesting results. Regarding instead the voice2song task, results were less encouraging. Even though some nice arrangements were produced, the model failed to properly and euphonically arrange the input voice lines. For this reason, **here** we limit to report some of the best produced samples, in the hope to improve the model greatly in the following months. As for baselines, initially we thought about comparing our results to three particularly notable works (Dhariwal et al.,

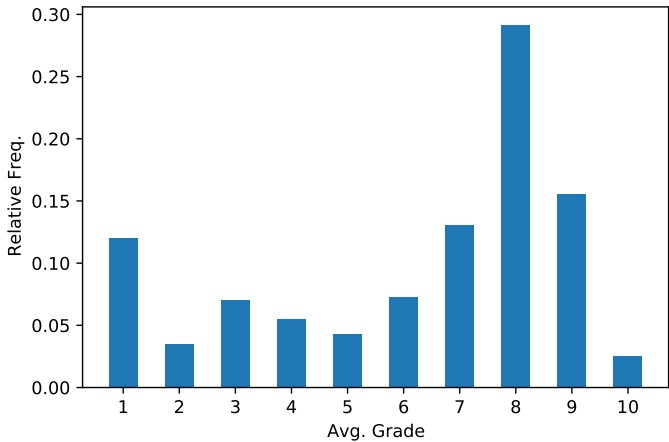

Figure 3: Grade distribution of generated drums samples

2020; Vasquez & Lewis, 2019; Mor et al., 2018), but after running some experiments we eventually realized that they could not be properly used for arrangement purposes. All three model produce very nice music samples, but none of them can take as input vocals or bass lines and produce a complementary arrangement. It is possible though that these models could be fine tuned to solve this new task. In addition, we replicated exactly the same experiments using Pix2Pix by Isola et al., a well known paired image-to-image architecture. Despite long training, results were very poor and quite unpleasant to listen to. Due to space concerns we do not report more details about this set of experiments.

Finally, with respect to the computational resources and time required to generate new arrangements, our approach shows several advantages, compared to auto-regressive models (Dhariwal et al., 2020). Since the output prediction can be fully parallelised, the inference time amounts to a forward pass and a Mel-spectrogram-waveform inverse conversion, whose duration depends on the input length, but it never exceeds few minutes. Indeed, it is worth noting that, at inference time, arbitrary long inputs can be processed and arranged.

## 5 Conclusions and Future Work

In this work, we presented a novel approach to automatically produce euphonic music arrangements starting from a voice line or a bass line. We applied Generative Adversarial Networks to real music pieces, treated as grayscale images (Mel-spectrograms). Given the novelty of the problem, we proposed a reasonable procedure to properly evaluate our model outputs. Notwithstanding the promising results, some critical issues need to be addressed before a more compelling architecture can be developed. First and foremost, a larger and cleaner dataset of source separated songs should be created. In fact, manually separated track always contain a big deal of noise. Moreover, the model architecture should be further improved to focus on longer dependencies and to take into account the actual degradation of high frequencies. Finally, a certain degree of interaction and randomness should be inserted to make the model less deterministic and to give creators some control over the sample generation. Our contribution is nonetheless a first step toward more realistic and useful automatic music arrangement systems and we believe that further significant steps could be made to reach the final goal of human-level automatic music arrangement production. Already now software like Melodyne (Neubäcker, 2011; Senior, 2009) delivers producers a powerful user interface to directly intervene on a spectrogram-based representation of audio signals to correct, perfect, reshape and restructure vocals, samples and recordings of all kinds. It is not unlikely that in the future artists and composers will start creating their music almost like they were drawing.

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
