# OpenReview forum: "Automatic Music Production Using Generative Adversarial Networks"
_ICLR.cc/2021/Conference — Reject_

### Official Review · AnonReviewer2 · 2020-10-28
**Interesting study, but needs more development**

**Rating:** 5
**Confidence:** 4

**Review:**

This paper describes an approach to what is often termed "automatic accompaniment" generation in music.
Given an input signal of one source (e.g., vocal or bass), the system is trained to generate one or more accompanying signals (drums, full arrangement).
The authors propose a CycleGAN model to learn transformations between source and accompaniment domains.
The model is trained on a combination of a large collection of automatically separate signals (FMA) and a small collection of isolated stem recordings (musdb).
The models for two tasks (bass->drum and vocal->full) were evaluated by a combination of human listener testing and automatic offline scoring, to somewhat mixed results.

Overall, I found this paper interesting and generally well-written.
The combination of subjective and offline evaluation was nice to see, and given the inherent difficulty of the problem, I don't consider the mixed results to be a total negative here.
That said, I do think there are areas in which this paper could be significantly improved, both in terms of experimental design and exposition.


The experiments presented here make use of both pre-separated stems (MusDB) and automatically separated signals produced by DEMUCS on the FMA dataset.
Given the size of available stem datasets, I understand the motivation for going this route.
However, I think there needs to be some quantitative evaluation of the impact of each part here, for several reasons:

1. DEMUCS is by no means perfect, and we should expect some bleed-through of the target signal (eg drums) into the separated signal (eg bass).  If this happens, the task becomes significantly easier than if the system was presented with clean stems.
2. We can't rely on previously reported BSS-EVAL metrics to give a sense of DEMUCS' performance on FMA for generating the training data.  The FMA dataset is quite different from MusDB in terms of production quality and instrumentation, and given the small size of MusDB, the reported metrics are almost certainly an over-estimate of quality we should expect on FMA.
3. It is not demonstrated that including the FMA data is necessary or beneficial for this task (though it's not unreasonable to expect that this is indeed true).  An experiment showing how the system performs if only trained end-to-end on musdb would make the existing results easier to interpret and place in context.


In terms of exposition, as stated above, I find the paper mostly clear and easy to follow.
However, many technical details are omitted that make it both impossible to reproduce and difficult to interpret.
The biggest omission here is the specific method for recovering the waveform from the generated Mel spectrograms.
Phase information is discarded early on in the process, but is critical to the perceptual quality of generated audio.
In listening to the included examples, it's pretty clear that there's a great deal of phase distortion in the results of both tasks.
(It's less perceptible in the drum synthesis task because the target signal does not generally consist of sustained tones, but it's still audible.)
This left me wondering how exactly the phase retrieval is done, and to a lesser extent, how the Mel spectrogram inversion is done.


Minor comments:

- The authors claim that the source separation model (DEMUCS) is time-equivariant (section 3.1), but I don't see how this is justified.  DEMUCS uses a U-net architecture with a bidirectional LSTM middle layer, which is not generally time-equivariant.

- Why are the spectrograms quantized to 256 values?  I agree that this probably doesn't introduce much distortion, but it seems unnecessary.  Point of clarification: are these spectrograms using linear magnitude or logarithmic (decibel) magnitude?  This decision would have a significant effect on how quantization is performed, but it's not clearly articulated in the paper.  Figure 1 suggests a log scaling, but does not provide details.  An equation would go a long way here.

- Is there any windowing applied in the short-time Fourier transform (eg Hann or Hamming)?  I would expect so based on the lack of transient artifacts in Figure 1, but it's not explicitly stated.  I ask because having listened to the provided examples, it sounds like there could be some modulation artifacts in the reconstruction that could be traced to the choice of window function.  Aside: if you're using an existing software package to implement your Mel spectrogram, it should be cited.

- I like the approach of mapping automatic scores to human judgments, but I'm confused as to why the targets were binarized.  Why not do an ordinary least squares or isotonic regression, that would discard less of the information?

---

> ### Author Response · Authors · 2020-11-23
> **The reviewer's attention to the signal processing part of our pipeline helped us to gain awareness and to strengthen the presentation of our contribution.**
>
> - DEMUCS is by no means perfect, and we should expect some bleed-through of the target signal (eg drums) into the separated signal (eg bass). If this happens, the task becomes significantly easier than if the system was presented with clean stems. $-> \textbf{Our answer}$ This has been clarified with a more in-depth explanation in the section 4.1
> - We can't rely on previously reported BSS-EVAL metrics to give a sense of DEMUCS' performance on FMA for generating the training data. The FMA dataset is quite different from MusDB in terms of production quality and instrumentation, and given the small size of MusDB, the reported metrics are almost certainly an over-estimate of quality we should expect on FMA. $-> \textbf{Our answer}$ This could surely be the case, and at first we were worried about possible contaminations. Manual inspection of separated drums and bass lines reassured us about the reliability of extracted signals. Both drums and bass contained very little or none bleed-through.
> - It is not demonstrated that including the FMA data is necessary or beneficial for this task (though it's not unreasonable to expect that this is indeed true). An experiment showing how the system performs if only trained end-to-end on musdb would make the existing results easier to interpret and place in context.  $-> \textbf{Our answer}$ Thank you for raising this issue! You are perfectly right, Unfortunately we were not able to conduct an in depth analysis on how much we can shrink the data set without losing quality in the outcomes. It has to be said that this is a very hot topic / open question in the deep learning community.
> - The biggest omission here is the specific method for recovering the waveform from the generated Mel spectrograms. Phase information is discarded early on in the process, but is critical to the perceptual quality of generated audio. In listening to the included examples, it's pretty clear that there's a great deal of phase distortion in the results of both tasks. (It's less perceptible in the drum synthesis task because the target signal does not generally consist of sustained tones, but it's still audible.) This left me wondering how exactly the phase retrieval is done, and to a lesser extent, how the Mel spectrogram inversion is done.  -$-> \textbf{Our answer}$ We agree with the reviewer, we added a detailed explanation of phase retrieval and time domain signal reconstruction in section 3.2
> - The authors claim that the source separation model (DEMUCS) is time-equivariant (section 3.1), but I don't see how this is justified. DEMUCS uses a U-net architecture with a bidirectional LSTM middle layer, which is not generally time-equivariant. $-> \textbf{Our answer}$ We added the following paragraph: it is worth adding that Demucs shows a nice property for a source separation model, namely it is time--equivariant, meaning that shift in the input mixture will cause a congruent shift in the output. The model does not naturally feature this property, but it is achieved through a workaround (randomized equivariant stabilization) as explained by the original authors.
> - Why are the spectrograms quantized to 256 values? I agree that this probably doesn't introduce much distortion, but it seems unnecessary. Point of clarification: are these spectrograms using linear magnitude or logarithmic (decibel) magnitude? This decision would have a significant effect on how quantization is performed, but it's not clearly articulated in the paper. Figure 1 suggests a log scaling, but does not provide details. An equation would go a long way here.  $-> \textbf{Our answer}$ We added details to clarify this point, by explaining how we transform a signal to a Mel-spectrogram representation, with some formulas. - Moreover, we changed the plot, which was indeed not very clear.
> - Is there any windowing applied in the short-time Fourier transform (eg Hann or Hamming)? I would expect so based on the lack of transient artifacts in Figure 1, but it's not explicitly stated. I ask because having listened to the provided examples, it sounds like there could be some modulation artifacts in the reconstruction that could be traced to the choice of window function. Aside: if you're using an existing software package to implement your Mel spectrogram, it should be cited.  $-> \textbf{Our answer}$ We added all required details in the paper
> - I like the approach of mapping automatic scores to human judgments, but I'm confused as to why the targets were binarized. Why not do an ordinary least squares or isotonic regression, that would discard less of the information? $-> \textbf{Our answer}$ Thank you for the observation. In principle we did not want to binarize targets, but after a thorough discussion with the evaluators we noticed that generated samples were either not acceptable for production purposes or acceptable. Within the these two groups the differences were not particularly marked.

---

### Official Review · AnonReviewer4 · 2020-10-29
**Promising directions but the study needs to be extended**

**Rating:** 4
**Confidence:** 3

**Review:**

In the paper, the authors adapt CycleGAN, a well-known model for unpaired image-to-image translation, to automatic music arrangement by treating MFCCs extracted from audio recordings as images. Also, the authors propose a novel evaluation metric, which learns how to rate generated audio from the ratings of (some) music experts. The authors make use of two large-scale datasets to train and evaluate the model on two scenarios, namely 1) generating drum accompaniment a given bass line, 2) generating arrangement given a voice line. They report promising results on the first task; however, the model is not as successful on the second (more challenging) task.

The problem is challenging, and meaningful solutions may bring innovative and creative solutions to music production. The literature is well-covered, with a few missing citations (see below). The approach is built upon existing work, and the experiments are conducted on two relevant, public datasets. On the other hand, the experimental code is not shared, and the dataset section lacks a few details to reproduce the findings easily.

Below are the shortcomings of the paper:

1. While adapting past music generation work for arrangement generation is not trivial, the authors could have still used variants of CycleGAN and other unpaired image-to-image translation models for comparison.
2. The sources are primarily limited to bass, drums, and vocals. I do not think the narrow scope is an issue on a paper focusing on an unexplored subject. On the contrary, the experiments could have more variety, e.g. drums2bass, bass&vocals2drums, and other combinations, so that we could examine which settings bring interesting and/or challenging outcomes in arrangement generation.
4. The evaluation and discussion could have more depth, e.g. inter-annotator agreement, the effect of source separation in the generated audio (separation errors, audible artifacts, ...)

The paper is novel in its application and brings promising results. However, the authors should extend the experiments, compare relevant models against each other, and discuss the results more in detail. Therefore, I would strongly encourage the authors to build upon their existing work and re-submit the revised paper to ICLR or another conference such as ISMIR.

Specific comments
=================

- As mentioned above, the authors should have added more "experimental settings." At least they should have included "generation of a bass line given the drums" (reverse of bass2drums) because 1) it would have allowed the readers to contrast the performance with bass2drums, 2) the task would be closer to the real-world use case (drums are typically the first to be recorded in a session followed by bass).

- The method works on music strictly with drums, bass and vocals, which is not mentioned until Section 3.4. This limitation/condition should be specified clearly and earlier in the Introduction and/or in Section 3.1.

- "Nevertheless, only raw audio representation can produce, at least in the long run, appealing results in view of music production for artistic and commercial purpose."

  Even if we restrict ourselves to popular music, this argument is too ambitious if not misleading. Many artists (performers, composers, conductors, etc.) are not only well fledged but - by profession - required to appreciate music by reading sheet music. Countless programmable interfaces and software, which make use of symbolic/hybrid music representations but do not generate raw audio directly, have been used extensively as part of music performances and production in a both artistic and commercial setting. While audio - without any doubt - is the essence of music, we can never disregard other representations.

- Citing the two papers below could improve the literature review:

  >Hawthorne, Stasyuk, Roberts, Simon, Huang, Dieleman, Elsen, Engel and Eck, "Enabling Factorized Piano Music Modeling and Generation with the MAESTRO Dataset", International Conference on Learning Representations, 2019. => similar to the authors' design decision, this paper uses a cheaper intermediate representation (music scores) for efficiency

  >Donahue et al. LakhNES: Improving multi-instrumental music generation with cross-domain pre-training => the paper involves mapping ("arranging") the instrumentation in MIDI files to NES sound channels.

- Please cite `FMA` and `MusDB18` datasets following the instructions in the respective online sources.

- Section 3.1. "While showing nice properties,"

  The authors only mention that Demucs solve audio source separation (for the data the authors use) and the algorithm is time equivariant. However, the text reads like the authors would like to state other properties as well. If there are others, they should be stated explicitly.

- Section 3.2.

  The authors should mention and cite the library they have used to extract MFCCs.

- Section 4.1 "we chose to select only pop music and its sub-genres for a total of approximately 10,000 songs"

  It would be beneficial to share IDs of the songs in the subset for reproducibility purposes. Also, the authors do not state whether they use the untrimmed or trimmed versions of the tracks in the FMA dataset, which is a crucial detail for model training as well as experimental reproducibility.

- The authors should state:

  1. number of songs used from the MusDB18 dataset (i.e. have they used both the train and test splits?)
  2. Total duration and number of samples in training, test and fine-tuning

- In the test set, instead, we chose only a few samples for each song due to the relative uniformity of its content: in other word, we expect our model to perform in similar ways on different parts of the same song.

  I find this assumption a bit unrealistic. In what sense, is the content uniform across the song? Is it uniformity in mixing, structure, arrangement, melody, tempo, or rhythm? Even if the authors use trimmed audio excerpts for training/testing, these characteristics can vary substantially within seconds (even if they use trimmed tracks).

  The authors should clearly state how they define content uniformity, provide a more informed argument around this assumption and experimentally show that the assumption holds for the test set.

- Section 4.2: "the result is somehow subjective thus different people may end up giving different or biased ratings based on their personal taste"

  The authors portrait subjectivity as unfavourable. However, - as a human construct - there are no objective, universal criteria for appreciating music. Likewise, the evaluation metric, which the authors are proposing, is based on the subjective responses from music experts. I think the justification needs rephrasing.

- Section 4.3: In the paper, the authors do not state the cultural background or the genre(s) of the focus of the music experts. The inter-agreement between the experts are not presented either. Due to lack of information and the small number of subjects, it is difficult to assess whether the (trained) evaluation metric has positive/negative/desired biases based on the experience, knowledge, personal taste etc. of the experts. Therefore, the claim about the proposed "metric correlating with human judgment" is a bit weak.

- What is the distribution of scores for bass and voice?

- How much do the artifacts (due to imperfections in source separation) affect the judgements?

Minor comments
==============

- Introduction, Paragraph 1: "allow artists and producers to easily manipulate recordings and create high quality songs directly from home."

  The phrasing somewhat disregards the music studios.

- Page 2, top row: "given a musical sample encoded in a two-dimensional time-frequency representation (known as Mel-spectrogram)"

  It reads like all two-dimensional time-frequency representations are called "Mel-spectrogram"s, instead of the authors using Mel-spectrograms, which is one type of two-dimensional time-frequency representations.

- The text should explain the relevance of the selected experimental settings to the music production: e.g. drums and bass are usually the first "sessions" to be recorded; a demo typically consists of the melodic prototype/idea with minimal accompaniment, which is later arranged by many collaborators...

- "Figure 1 shows a Mel-spectrogram example, a visual representation of a spectrum, where the x axis represents time, the y axis represents the Mel bins of frequencies and the third gray tone axis represents the intensity of the sound measured in decibel (Briot et al., 2020)."

  I do not understand what the authors mean by "third gray tone axis." Is it because the MFCCs are treated as a single channel image, hence "gray"? If yes, it is better to state that the "MFCCs are treated as a single channel image" without resorting to image processing jargon.

- "Mel-frequency cepstral coefficients are the dominant features used in speech recognition, as well as in some music modeling tasks (Logan & Robinson, 2001)"

  It may be better to introduce this sentence earlier in the paragraph.

- Section 3.4: "On the one hand, ... On the other hand"

  It might be easier to read if the setting is enumerated for readability.

- Section 4.1: "To train and test our model We decide"

  Lowercase "We" -> "we"

- MusDB18 URL is broken

- Section 4.3: "Time: a rating from 1 to 10 of whether the produced drums and arrangements are on time the the bass and voice lines"

  Double "the the" -> "with the"

---

> ### Author Response · Authors · 2020-11-23
> **Thanks to the reviewer's clever remarks we were able to address several weaknesses of our work.**
>
> - the experimental code is not shared, and the dataset section lacks a few details to reproduce the findings easily.  $-> \textbf{Our answer}$: The pipeline has been implemented in PyTorch and all the code will be released upon acceptance to promote reproducibility, we also added all the necessary details to the paper.
> - The authors could have still used variants of CycleGAN... $-> \textbf{Our answer}$: When writing the paper, due to space concerns and the irrelevance of results, we decided to leave out experiments conducted with the Pix2Pix architecture.  We made this point clear in section 4.5
> - The sources are primarily limited to bass, drums, and vocals... $-> \textbf{Our answer}$: Thanks to the cycle consistency property of CycleGAN, drums2bass samples were automatically generated as well. Nonetheless, we did not add this task because creating a bass line from scratch is outside of our objective. Given drums, the system would not have enough harmonic information to generate a bass accompaniment. This is more of a generative task.
> - The evaluation and discussion could have more depth... $-> \textbf{Our answer}$: In table 1 we added the correlation matrix for all 4 annotators. We used the Pearson correlation because of the continuous nature of the averaged scores.
> - The method works on music strictly with drums, bass, and vocals... $-> \textbf{Our answer}$: We already specified this condition in the Introduction actually. We recalled this aspect in Section 3.4, where we describe more in detail the case study presented in the Introduction.
> - "Nevertheless, only raw audio representation can produce, at least in the long run, appealing results in view of music production for artistic and commercial purpose." ... $-> \textbf{Our answer}$: The statement was rephrased.
> Citing the two papers below could improve the literature review... $-> \textbf{Our answer}$: these references were added.
> - Please cite FMA and MusDB18 datasets ... $-> \textbf{Our answer}$: done!
> - The authors only mention that Demucs ... is time equivariant... $-> \textbf{Our answer}$: There may have been a misunderstanding, the authors would not like to state or observe other properties. The text in the relevant section has been changed to make it clearer.
> - The authors should mention and cite the library they have used to extract MFCCs. $-> \textbf{Our answer}$: done!
> - It would be beneficial to share IDs of the songs in the subset for reproducibility purposes... $-> \textbf{Our answer}$: done!
> - The authors should state the number of songs used from the MusDB18 dataset $-> \textbf{Our answer}$: We add these pieces of information in the data set section.
> - In the test set, instead, we chose only a few samples for each song due to the relative uniformity of its content... $-> \textbf{Our answer}$: sorry, we did not explain ourselves well: we meant that, in the evaluation pipeline, we only converted a random selection of samples from each test song, instead of the whole song.
> - The authors portrait subjectivity as unfavorable... $-> \textbf{Our answer}$: We rephrased the justification at the beginning of section 4.3.
> - Section 4.3: In the paper, the authors do not state the cultural background or the genre(s) of the focus of the music experts... $-> \textbf{Our answer}$: In table 1 we added the correlation matrix for all 4 annotators.
> - What is the distribution of scores for bass and voice? -> They were all considered optimal because the came from high-quality productions.
> - How much do the artifacts (due to imperfections in source separation) affect the judgments?  $-> \textbf{Our answer}$: This is very hard to say and quantify. In the future, we plan to dig deeper into the data quality requirement. For the bass2drums task imperfections in source, separation were virtually absent.
> - Introduction, Paragraph 1...The phrasing somewhat disregards the music studios.  $-> \textbf{Our answer}$: We agree with the reviewer, this statement was too sharp; because of this, we changed it with “provide a wide set of tools to manipulate recordings and simplify the composition process for artists and producers”.
> - Page 2, top row... It reads like all two-dimensional time-frequency representations are called "Mel-spectrogram"s... $-> \textbf{Our answer}$: We agree with the reviewer, the statement is ambiguous; because of this, we changed “(known as Mel-spectrogram)” to “(in particular, we opted for the Mel-spectrogram time-frequency representation)”.
> - The text should explain the relevance of the selected experimental settings...  $-> \textbf{Our answer}$: We added a couple of lines to explain the relevance of the selected experimental settings in section 3.4
> - "Figure 1 shows a Mel-spectrogram example...  $-> \textbf{Our answer}$: We rephrased as “which is treated as a single-channel image, representing the sound intensity with respect to time - x-axis - and frequency  - y-axis”.

---

### Official Review · AnonReviewer1 · 2020-11-02
**Review of paper on Automatic Accompaniment in the Frequency Domain Using CycleGAN**

**Rating:** 2
**Confidence:** 5

**Review:**

This paper proposes a method for automatically generating accompaniments using Mel-spectrograms as inputs to a CycleGAN. Overall I think the paper requires significant revision and additional work before it can be accepted as a conference publication.

Title:

-The title is misleading. The title claims that the proposed model is for "Automatic Music Production". However the actual task considered is more restrictive. The authors propose a model for automatic accompaniment. Music Production involves many other tasks like mixing, mastering and so on, none of which are a part  of this study. The title should therefore be updated to be more specific.

Abstract:

-"Despite consistent demands from producers and artists...": I think this sentence should be rephrased to motivate the need for automatic accompaniment from a different angle. If not, the authors should present some justification for the demand for this technology from artists and producers.

-"Automatic music arrangement from raw audio in the frequency domain": why not simply say automatic music arrangement/accompaniment in the Mel-frequency domain? I find the raw audio part of the description unnecessary and confusing.

-The authors claim that the they are the first to treat music audio as images and then apply techniques from computer vision. However, treating spectrograms as images is the current standard for many MIR tasks like music transcription, chord recognition and so on e.g. "An end-to-end Neural Network for Automatic Music Transcription": https://ieeexplore.ieee.org/abstract/document/7416164/. There are hundreds of other publications that are similar to this approach.

Introduction:

-The authors claim that automatic accompaniment in the waveform/frequency domain has many advantages. However they fail to motivate the short-comings of this approach. Namely the lack of source separated training data and the extreme difficulty in source separation for music recordings. It  would also be useful to cite a review paper or some of the many publications on automatic accompaniment generation in the symbolic domain so that the reader can find references to this problem which has an extensive literature already.

-The authors mention that they use the Demucs algorithm for source separation. However they do not provide any details whatsoever about this approach, especially the downsides. A quick scan of the paper reveals that the algorithm introduces severe artefacts under various conditions.

-The authors mention the low-computational cost of their proposed method, however they do not satisfactorily quantify this claim. Firstly, is computational cost an issue? Does this algorithm have to run on a mobile device? Will it be run in a streaming setting? These questions are not answered in the paper.

Related Works:

-The authors cite many papers on music generation in the waveform domain however they do not cite any of the extensive literature on music generation in the symbolic domain. This literature is extremely relevant to the work presented in this paper.

-"Nevertheless, only raw audio representation can produce, at least in the long run, appealing results in view of music production for artistic and commercial purposes." Why is this the case? Why is generating music in the symbolic domain and then using state-of-the-art synthesisers not an appealing direction? This point isn't made clear in the paper.

Method:

-There are no details provided about the Demucs algorithm used to separate the source training data into various channels like vocal, bass, drums etc. How big was the model? Did the authors train the model themselves? Did they use a pre-trained model? Were there any artefacts present in the source separated tracks? Are there any downsides to this algorithm? Are there any alternatives to this algorithm? Do the artefacts not interfere with the  downstream task?

-A reference/citation about the Mel scale would be useful.

-There are no details about the CycleGAN used in the paper. How big is the model? What is the architecture? How was it trained? What flavour of gradient descent was used for training? What are the hyper-parameters? Was the model trained on a single GPU?

Experiments:

-How was the subset of pop music selected? How was the metadata filtered to obtain the 10000 tracks used for training? If the filtering algorithm cannot be outlined, then it would be useful to provide a list of the 10000 tracks used for training, for the purpose of reproducibility.

-How did the authors arrive on the 4 attributes quality, euphony, coherence and intelligibility? Is there some theory that suggests that these 4 attributes would be useful in determining whether the accompaniment is somehow good? These attributes have been presented without justifications and citations.

-The features (STOI, FID) used to compare the automatically generated accompaniment have also been presented without much justification. Why is it that these features  are an adequate representation of the generated audio?

-I found the description of the grades and the subsequent comparison in Figure 3 difficult to follow. I think the description needs to be significantly more rigorous.

---

> ### Author Response · Authors · 2020-11-23
> **The reviewer made several very thorough comments. We tried to address all of them and we think that, thanks to the reviewer's suggestions, the paper is much stronger now.**
>
> Title + Abstract
>
> - The title is misleading... $-> \textbf{Our answer}$: we decided to keep it as it is for two reasons: first, quite often producers do not mix nor master songs since they consider these tasks as post-production. Surely, many others see music production as the whole process, but the issue is debatable. Second, even though we agree with you that the challenge we tackle is more restrictive, we wanted to contextualize it as part of a long-term effort towards completely automatic music production.
> - "Despite consistent demands from producers and artists..." $-> \textbf{Our answer}$:  we rephrase this sentence highlighting the reasons why the automatic arrangement generation is so important for producers, artists, and companies.
> - "Automatic music arrangement from raw audio in the frequency domain" $-> \textbf{Our answer}$: we changed the sentence to “automatic music accompaniment in the Mel-frequency domain”.
> - The authors claim that they are the first to treat music audio as images... $-> \textbf{Our answer}$: the main innovations of our contribution are now stated more clearly at the end of the introduction. Moreover, we added the proposed reference.
>
> Introduction
>
> - The authors claim that automatic accompaniment in the waveform/frequency domain has many advantages... $-> \textbf{Our answer}$:  for more details on the limitations and advantages of the waveform/frequency domain and the main results for the symbolic domain, we modified the “Related works”.
> - The authors mention that they use the Demucs algorithm for source separation...  $-> \textbf{Our answer}$: we added details!
> - The authors mention the low-computational cost of their proposed method, however, they do not satisfactorily quantify this claim... $-> \textbf{Our answer}$:  the claim has been rectified. By low-computational cost, we were referring to the fact that our method is fully parallelizable and does not need to go through the lengthier procedure caused by autoregressive mechanisms used by comparable models in the field like the one cited in the paper.
>
> Related works
>
> - The authors do not cite any of the extensive literature on music generation in the symbolic domain... $-> \textbf{Our answer}$:  we added to the “Related Works” section many relevant papers dealing with symbolic representation. We did not in the beginning because the two research threads ofter are seen as parallel.
> - "Nevertheless, only raw audio representation can produce, at least in the long run, appealing results..." $-> \textbf{Our answer}$: We added the following statement to the related section: another very promising approach would be to work with symbolic music and then use state-of-the-art synthesizers to produce sounds. MIDI, music sheets, and piano rolls, however, are not always easy to find or produce. Moreover, many musicians and artists can not real music and would be more comfortable to work in a less formalized setting. Finally, state-of-the-art synthesizers, although increasingly indistinguishable from live recordings, can not yet reproduce the infinite nuances of real voices and instruments. Conversely, raw audio representation could be more appealing for some creators given its flexibility and little music competence required.
>
> Method
>
> - There are no details provided about the Demucs algorithm... $-> \textbf{Our answer}$:  We added relevant details under section “Source Separation for Music”
> - A reference/citation about the Mel scale... $-> \textbf{Our answer}$:  Added relevant citation.
> - There are no details about the CycleGAN used in the paper...  $-> \textbf{Our answer}$:  A dedicated section (4.2) was added to answer all of the questions.
>
> Experiments
>
> - How was the subset of pop music selected? $-> \textbf{Our answer}$: Thank you for the observation. You can now find the following line in the datasets section
> - How did the authors arrive at the 4 attributes of quality, euphony, coherence, and intelligibility? $-> \textbf{Our answer}$: To clarify this point we added: the choice fell on these four aspects after a thorough with the evaluators. We asked them to list and describe the most relevant dimensions in evaluating the quality of a piece of pop-rock music.
> - The features (STOI, FID) $-> \textbf{Our answer}$: To be honest, we do not assume that these features adequately represent the generated audio. Instead, we tried to extract/engineer two sets of features that could predict human judgment. After several attempts, we ended up using a modified version of STOI and FID.
> - I found the description of the grades and the subsequent comparison in Figure 3 difficult to follow. I think the description needs to be significantly more rigorous. $-> \textbf{Our answer}$: We rephrased the section to better highlight our intentions and how we proceeded. Nonetheless, it is important to stress that these descriptions stem from an ex-post analysis of the results and were not given as guidelines to raters.

---

### Decision · Program_Chairs · 2021-01-07
**Final Decision**

**Decision:**

Reject

**Comment:**

All Reviewers point out that the paper, although having some strong points, does not meet the bar for a highly-selective machine learning conference like ICLR. Hence, my recommendation is to REJECT the paper. As a brief summary, I highlight below some pros and cons that arose during the review and meta-review processes.

Pros:
- Well-written paper.
- Ambitious task.
- Code will be released.

Cons:
- Unclear task terminology (music production; misleading title).
- Mixed results.
- Experimental design could be improved.
- Exposition could be improved (technical details missing).
- Lack of comparison (for instance with other CycleGAN variants; more experimental setups).
- Lack of discussion on the use of a source algorithm for pre-processing data.